# Effect of Morphological Characteristics and Biomineralization of 3D-Printed Gelatin/Hyaluronic Acid/Hydroxyapatite Composite Scaffolds on Bone Tissue Regeneration

**DOI:** 10.3390/ijms22136794

**Published:** 2021-06-24

**Authors:** Jae-Woo Kim, Yoon-Soo Han, Hyun-Mee Lee, Jin-Kyung Kim, Young-Jin Kim

**Affiliations:** 1Department of Biomedical Engineering, Daegu Catholic University, Gyeongsan 38430, Korea; wodnek1122@hanmail.net; 2Department of Chemical Engineering, Daegu Catholic University, Gyeongsan 38430, Korea; yshancu@cu.ac.kr; 3Department of Optometry and Vision Science, Daegu Catholic University, Gyeongsan 38430, Korea; hmlee@cu.ac.kr; 4Department of Biomedical Science, Daegu Catholic University, Gyeongsan 38430, Korea; toto0818@cu.ac.kr

**Keywords:** bone regeneration, 3D printing, composite scaffold, morphological characteristic, biomineralization

## Abstract

The use of porous three-dimensional (3D) composite scaffolds has attracted great attention in bone tissue engineering applications because they closely simulate the major features of the natural extracellular matrix (ECM) of bone. This study aimed to prepare biomimetic composite scaffolds via a simple 3D printing of gelatin/hyaluronic acid (HA)/hydroxyapatite (HAp) and subsequent biomineralization for improved bone tissue regeneration. The resulting scaffolds exhibited uniform structure and homogeneous pore distribution. In addition, the microstructures of the composite scaffolds showed an ECM-mimetic structure with a wrinkled internal surface and a porous hierarchical architecture. The results of bioactivity assays proved that the morphological characteristics and biomineralization of the composite scaffolds influenced cell proliferation and osteogenic differentiation. In particular, the biomineralized gelatin/HA/HAp composite scaffolds with double-layer staggered orthogonal (GEHA20-ZZS) and double-layer alternative structure (GEHA20-45S) showed higher bioactivity than other scaffolds. According to these results, biomineralization has a great influence on the biological activity of cells. Hence, the biomineralized composite scaffolds can be used as new bone scaffolds in bone regeneration.

## 1. Introduction

The reconstruction of damaged bone defects originating from tumors, trauma, infections, and congenital malformations is a complex biological process that requires osteoconductive scaffolds, osteogenic precursor cells, and osteoinductive growth factors, which also needs precise control of bleeding disorders by congenital afibrinogenemia at the defect site [1,2]. Although the conventional method for bone repair is the use of autologous bone grafts, alternative materials are necessary for the repair of large bone defects because of autograft supply limitations, the risk of rejection, and donor site morbidities [1,2,3,4]. Thus, tissue-engineered three-dimensional (3D) scaffolds are currently recognized as ideal substitutes for autologous bone grafts because of their biocompatibility and osteoconductivity. These scaffolds should mimic the physical and chemical properties of the extracellular matrix (ECM) to promote bone regeneration, implying that they should provide a conducive microenvironment for the selected cells [5,6].

Several fabrication methods, such as freeze drying, electrospinning, and double emulsion methods, have been explored for preparing porous scaffolds for bone tissue engineering [7,8,9]. Nevertheless, these methods do not provide accurate control over the porosity, pore size, and spatial distribution of pores. As an advanced fabrication technology, 3D printing has recently attracted great attention in the biomedical field because of its versatility, ease of use, and precise control of a customized shape with unique architecture [4,6,10]. These advantages enable the successful fabrication of bone scaffolds with a predetermined pore structure, where pore structure (i.e., porosity and pore size) is a critical parameter to determine vascularization and osteointegration during bone regeneration.

3D printing is a versatile technology for creating precise shapes and macro/microporous structures in biomaterials, such as ceramics, polymers, and composites, for bone tissue engineering [3,6,11,12]. Among various biomaterials, natural biopolymers, such as gelatin, silk fibroin, and bacterial cellulose, exhibit excellent biocompatibility and superior cell recognition ability, but their mechanical properties are insufficient for bone tissue engineering application [3,12,13]. In addition, hydroxyapatite (HAp), the main inorganic component of hard tissue in natural bone, has been widely used in bone tissue engineering because of its good bioactivity [6,8,12]. However, HAp has limitations in its applications because of issues such as low mechanical strength, brittleness, and lack of machinability.

Several studies have explored the fabrication of composite scaffolds to promote proliferation and differentiation of bone cells and improve the mechanical properties of composite scaffolds [12,13,14]. Most composite scaffolds have been fabricated using the direct blending method of HAp nanoparticles with biodegradable polymer and deposition from simulated body fluid (SBF) [12,15]. Although direct blending can improve the mechanical properties of composite scaffolds, control of the homogenous distribution of HAp nanoparticles in the polymer matrices is difficult. In addition, SBF deposition produces the composite scaffolds with only the external surface coated with low-crystalline apatite. These composite scaffolds are not effective for cell proliferation in long-term culture because of the fast dissolution of the apatite layer [16].

As a leading manufacturing technology, 3D printing can fabricate highly ordered pore structures and customized shapes, which have a wide range of applications, including bone repair scaffolds [11,12]. The bioactivity of bone scaffolds significantly depends on porosity and pore sizes, which are crucial for vascularization, cell migration into the scaffold, and nutrient supply to developing tissue. In addition, these characteristics of pore structure also influence the surface area and mechanical properties of scaffolds [17].

Therefore, we developed a new method, by combining 3D printing and biomineralization, for fabricating biomimetic 3D composite scaffolds with precisely controlled architecture to support bone tissue regeneration. The prepared composite scaffolds were covered with a high concentration of apatite crystals not only on the inner side but also on the external surface for potential use in long-term applications. We systematically investigated the effect of biomineralization on the physicochemical and mechanical properties of the fabricated composite scaffolds. In addition, we also evaluated the effect of geometrical configuration change of the fabricated composite scaffolds on morphological characteristics, such as pore size and surface area. The biological activity of the composite scaffolds was investigated through in vitro studies to verify their availability for bone tissue regeneration.

## 2. Results and Discussion

### 2.1. Fabrication and Characterization of the 3D Composite Scaffolds

An ideal scaffold for bone tissue regeneration should promote proliferation and differentiation of bone-forming cells, resulting in a new bone formation [15]. In this study, the 3D gelatin/HA/HAp composite scaffolds were fabricated using 3D printing with a low-temperature freezing system by varying several conditions, such as the solution viscosity, gelation temperature, and plate temperature. Among them, the solution viscosity was found to be most critical to the 3D printing [18]. It was reported that the aqueous gelatin solution showed thermo-responsive behavior without chemical crosslinking [19]. We investigated the optimal condition of the gelatin/HA solution for 3D printing by examining the rheological behavior of the gelatin/HA solution at various temperatures. A 6.3% (*w*/*v*) gelatin/HA solution exhibited the sol state at 25 °C and did not maintain the 3D structural frame at this temperature. However, the viscosity of the solution significantly increased below 23 °C, as shown in Figure 1a. In addition, the gel point of gelatin/HA solution was observed at 22.4 °C (Figure 1b). Resultingly, the structural frame was maintained during 3D printing by controlling the temperature of the cryogenic plate.

We first fabricated the 3D double-layer orthogonal scaffolds containing 6.3% (*w*/*v*) gelatin/HA and different amounts of HAp nanoparticles using a 3D printing method. The contents of HAp nanoparticles were 0 (GEHA0), 10 (GEHA10), 20 (GEHA20), and 40 wt% (GEHA40) with respect to the weight of the gelatin and HA. All scaffolds exhibited uniform structure and homogeneous pore distribution (Appendix A). A higher amount of HAp nanoparticles was observed on the surface of GEHA40. In addition, the mechanical property of the composite scaffolds increased with increasing the content of HAp nanoparticles (Appendix A). However, GEHA20 and GEHA40 exhibited almost the same mechanical strength and chemical structure. Therefore, the content of HAp nanoparticles in the composite scaffolds was fixed at 20 wt%.

Next, we designed the final object models using 3D modeling CAD software to evaluate the effect of morphological characteristics on the biological activity of the 3D composite scaffolds (Figure 2a). The geometrical configurations of double-layer orthogonal (GEHA20), double-layer staggered orthogonal (GEHA20-ZZ), and double-layer alternative scaffolds (GEHA20-45) were designed and used for 3D printing of the composite scaffolds. The final GEHA20, GEHA20-ZZ, and GEHA20-45 models were produced by placing two equal layers at a 90° rotation, by offsetting consecutive layers with 0°/90° strand orientation, and by rotating layers on a 0°/45°/90°/135° configuration, respectively. In addition, the surface area of the final models was calculated using 3D modeling CAD software, which indicated that GEHA20-ZZ had the highest surface area among the models (Figure 2b).

The 3D composite scaffolds were fabricated via a 3D printing of gelatin/HAp and subsequent biomineralization. The surface and cross-sectional morphologies of the cross-linked scaffolds after 3D printing using predesigned models were observed via scanning electron microscopy (SEM). The macromorphologies of the resulting scaffolds differed, as shown in the SEM images in Figure 3. However, all prepared scaffolds exhibited uniform structure and homogeneous pore distribution, and the microstructures of the three types of scaffolds showed an ECM-mimetic structure with a wrinkled internal surface and porous hierarchical architecture. In addition, HAp nanoparticles were observed on the surface of the scaffolds. Furthermore, the crosslinking degrees of the scaffolds were calculated from the moles of free amine groups per gram of gelatin according to the previous report [20]. All samples exhibited almost the same crosslinking degree, for example, 20.0 ± 1.8% for GEHA20, 22.1 ± 1.9% for GEHA-ZZ, and 21.5 ± 1.5% for GEHA20-45.

After biomineralization, the macrostructures of the composite scaffolds (i.e., GEHA20S, GEHA20-ZZS, and GEHA20-45S) were unchanged. In comparison, the deposition of inorganic apatite crystals entirely changed the microstructures of the composite scaffolds, meaning that small clusters of apatite crystals formed on individual surfaces of the composite scaffolds (Figure 4). This proved that biomineralization mainly occurred at the surface of the scaffolds’ strands, leading to the formation of bead-shaped clusters of nanoapatite crystals on the surface of the composite scaffolds. These clusters can accelerate the biomineralization process to form more nanoapatite crystals.

The surface roughness of scaffolds can influence cell attachment behavior, implying that the increase in surface roughness enhances cell attachment [21]. Therefore, the effect of biomineralization on the surface roughness of the scaffolds was examined via atomic force microscopy (AFM). According to AFM analysis results, biomineralization slightly affected the surface roughness of scaffolds (Figure 5a). The arithmetic average roughness (Ra) was calculated for the solid regions of the scaffolds, showing that Ra slightly increased from 2.3 nm (GEHA20) to 13.1 nm (GEHA20S) after biomineralization; this is caused due of the formation of bead-shaped clusters on the surface of the scaffolds. The Ra values of the other scaffolds also slightly increased from 2.1 nm (GEHA20-ZZ) to 13.2 nm (GEHA20-ZZS) and from 2.5 nm (GEHA20-45) to 13.4 nm (GEHA20-45S) after biomineralization. In addition, after biomineralization, the surface of the scaffolds was covered with nanoapatite crystals.

In bone tissue engineering, the function of scaffolds is to provide a 3D spatial and temporal structure to direct cell attachment, proliferation, differentiation, and bone tissue formation [1,4,22]. From this viewpoint, open porous architecture with a suitable pore size ranging from 100 to 300 µm is required to facilitate mass transportation of nutrients and vascularization. As shown in Figure 5b, the pore size of GEHA20S was higher than that of others: 449 ± 33 µm for GEHA20S, 233 ± 35 µm for GEHA20-ZZS, and 291 ± 31 µm for GEHA20-45S. Based on this result, GEHA20-ZZS and GEHA20-45S have suitable pore sizes for bone tissue engineering.

Fourier transform infrared spectroscopy (FTIR) was used to analyze the chemical structure of the 3D composite scaffolds by identifying the functional groups of the scaffolds. The spectra of all samples before biomineralization exhibited characteristic bands at 3276, 1629, and 1536 cm^−1^ corresponding to the N–H stretching vibration mode, C=O stretching vibration, and the coupling of the N–H bending vibration and C–N stretching vibration modes in gelatin, respectively (Figure 6a). A free N–H stretching vibration appeared in the range of 3400–3440 cm^−1^. This peak shifted to lower frequencies, ~3300 cm^−1^, after introducing the N–H groups of proteins into hydrogen bonding [23]. In addition, the absorption band that occurred at 1234 cm^−1^ was associated with the C–O–C stretching vibration in HA. Moreover, the absorption peaks at 1026, 600, and 560 cm^−1^ were assigned to the P–O stretching vibration and O–P–O bending vibration modes, respectively [16]. However, changes in characteristic bands after biomineralization were not observed, except for the nano apatite increase in band intensity at 1026 cm^−1^. The surface elemental composition of the composite scaffolds was examined using EDX to verify the fabrication of the composite scaffolds. The composite scaffolds before and after biomineralization exhibited only five peaks, related to C, N, O, P, and Ca, even though the intensity of the peaks assigned to Ca and P increased after biomineralization (Figure 6b).

The crystalline phases of the composite scaffolds were measured via X-ray diffraction (XRD), as shown in Figure 6c. The composite scaffolds before biomineralization exhibited broad diffraction peaks, which were typical XRD patterns of amorphous characteristics of gelatin. By contrast, the composite scaffolds after biomineralization revealed typical diffraction patterns that were consistent with the JCPDS database (JCDPDS 09-0432) for the HA crystalline phase. The characteristic peaks at 25.9°, 28.4°, 29.1°, 31.7°, 32.8°, 34.2°, 40.0°, 46.8°, 48.2°, 49.4°, and 53.5° were indexed to the (002), (102), (210), (211), (300), (202), (310), (222), (312), (213), and (004) planes of the HA crystal, respectively [16,24]. The scaffold should be sufficiently strong to withstand forces during new bone tissue regeneration. Therefore, the change in the mechanical properties of the composite scaffolds was measured based on the compressive strength test. As shown in Figure 6d, the compressive strength of the composite scaffolds slightly increased after biomineralization. In addition, the effect of morphology on the mechanical properties of the composite scaffolds was assessed, implying that double layer staggered orthogonal scaffolds (GEHA20-ZZ and GEHA20-ZZS) demonstrated higher compressive strength compared with the other scaffolds. Based on these results, the composite scaffolds fabricated in this study exhibited similar compressive strength to general gelatin-based composite scaffolds for bone tissue engineering [25,26].

### 2.2. Cell Attachment and Proliferation on the 3D Composite Scaffolds

Scaffolds in bone tissue engineering should promote cell growth and physiological function and maintain normal states of cell differentiation. Therefore, the bioactivity of the composite scaffolds was investigated to evaluate the potential of the scaffolds for bone tissue regeneration by evaluating the attachment and proliferation of human bone marrow-derived mesenchymal stem cells (hBMSCs) on the composite scaffolds. The proliferation behavior of hBMSCs cultured on the composite scaffolds was evaluated via the MTT assay, with higher cell viability meaning better cytocompatibility.

Figure 7a shows that the number of hBMSCs on all tested samples increased with the culture time. Changes in morphological characteristics of the composite scaffolds slightly influenced cell viability. The double-layer orthogonal-type scaffolds (GEHA20 and GEHA20S) with low surface area and high pore size did not effectively promote cell proliferation compared with double-layer staggered orthogonal-type (GEHA20-ZZ and GEHA20-ZZS) and double-layer alternative-type scaffolds (GEHA20-45 and GEHA20-45S). In addition, biomineralization significantly affected cell proliferation, indicating that the number of hBMSCs increased faster on the biomineralized composite scaffolds than on untreated scaffolds. Therefore, among the composite scaffolds, the GEHA20-ZZS and GEHA20-45S scaffolds showed the highest proliferation of hBMSCs at all tested time points.

Confocal laser scanning microscopy (CLSM) was used to investigate the attachment and morphology of cells by analyzing the cytoskeleton to examine cell growth on the composite scaffolds. The cultured hBMSCs were subjected to previous immunocytochemical procedures with phalloidin-TRITC (red) to stain cytoskeletal F-actin and DAPI (blue) to counterstain the nuclei, respectively. The resulting CLSM images of hBMSCs on the surface of the composite scaffolds cultured for 7 days showed excellent cell attachment, with cells exhibiting a normal spindle shape (Figure 7b). These spindle-shaped hBMSCs exhibited a well-stretched morphology characterized by elongated actin filaments, which were uniformly distributed over the surfaces of strands in the scaffolds. The number of stained hBMSCs on the double-layer staggered orthogonal-type and double-layer alternative-type scaffolds was higher than on the double-layer orthogonal-type scaffolds. Biomineralization, in particular, stimulated hBMSCs proliferation, indicating that the number of cells on the biomineralized composite scaffolds was greater than on the unmineralized scaffolds. This is in good agreement with the results of the MTT assay.

### 2.3. Cell Differentiation on the 3D Composite Scaffolds

One of the primary functions of hBMSCs is their ability to differentiate into various lineages, of which osteogenic differentiation is the most desirable for a bone implant. Alkaline phosphatase (ALP) activity is commonly used as an early indicator of osteogenic differentiation of cells because ALP expression affects the formation of bone mineral [13,16]. As displayed in Figure 8a, a highly significant increase in ALP activity in hBMSCs cultured on GEHA20-ZZS and GEHA20-45S was observed on day 14 compared with the other scaffolds. The ALP activity of osteogenic differentiating cells is affected by cell confluence and interaction with osteoinductive elements on the scaffolds [24]. Therefore, the biomineralized surface of the composite scaffolds enormously stimulated the expression of ALP activity, implying that GEHA20-ZZS and GEHA20-45S had the highest APL activity as expected. These results suggest that the morphological characteristics and biomineralization of the composite scaffolds significantly influence the osteogenic differentiation of hBMSCs.

Bone remodeling involves a signaling/coupling process by both osteoprotegerin (OPG) and nuclear factor kappa-B ligand (RANKL) markers, balancing bone formation and bone resorption [27]. Balancing the ratio of OPG to RANKL is considered a key mechanism in the regulation of bone remodeling. OPG acts as a decoy receptor and antagonistically binds to RANKL. Therefore, it can inhibit osteoclastogenesis and induce apoptosis in preexisting osteoclasts, resulting in a dose-dependent downregulation of bone resorption [28]. If OPG is higher, bone formation dominates; conversely, if RANKL is higher, bone resorption occurs. As shown in Figure 8b–d, the biomineralized composite scaffolds showed increased OPG and decreased RANKL expression compared with the unmineralized scaffolds. In particular, GEHA20-ZZS and GEHA20-45S exhibited a higher OPG/RANKL ratio than the other scaffolds. These increased OPG and decreased RANKL expression may increase bone formation and reduce bone resorption.

In the process of osteogenic differentiation, ALP is highly expressed in the early stage, and then special ECM proteins, such as collagen type 1 (COL1), runt-related gene 2 (RUNX2), and osteopontin (OPN), are secreted as typical osteogenic differentiation markers to initiate the mineralization process [16,21]. The expressions of COL1, RUNX2, and OPN on day 14 were analyzed via quantitative real-time reverse transcription–polymerase chain reaction (qRT-PCR), which was higher on the biomineralized composite scaffolds than on the unmineralized scaffolds, as displayed in Figure 9. According to this result, the biomineralization of scaffolds can considerably promote osteogenic differentiation of hBMSCs. However, morphological characteristics did not significantly influence osteogenic differentiation, as suggested by a minor change in the relative expression level of protein markers, even though the change was statistically significant. According to the results of the cell assays, biomineralization has a significant effect on not only short-term behaviors of cells, such as attachment and spreading, but also on the long-term behaviors, such as differentiation, whereas morphological characteristics only have a minor influence on the behaviors of cells.

## 3. Materials and Methods

### 3.1. Materials

Hyaluronic acid (HA) sodium salt from *Streptococcus equi* (8–15 kDa), HAp nanoparticle, *N*-(3-dimethylaminopropyl)-*N′*-ethylcarbodiimide hydrochloride (EDC), *N*-hydroxysuccinimide (NHS), 2,4,6-trinitro-benzensulfonic acid (TNBS), dexamethasone, β-glycerophosphate disodium salt hydrate, L-ascorbic acid, and 3-(4,5-dimethyl-2-thiazolyl)-2,5-diphenyl-2H-tetrazolium bromide (MTT) were obtained from Sigma–Aldrich (St. Louis, MO, USA). Gelatin from porcine skin (250 Bloom) was obtained from Geltech (Busan, Korea). Human bone marrow-derived mesenchymal stem cells (hBMSCs) were obtained from the American Type Culture Collection (ATCC, Manassas, VA, USA). Minimum essential medium alpha (MEM-α), fetal bovine serum (FBS), penicillin–streptomycin, and Dulbecco’s phosphate-buffered saline (DPBS, pH 7.4) were purchased from Cytiva (Logan, UT, USA). The Actin Cytoskeleton and Focal Adhesion Staining Kit was purchased from Merck Millipore (Burlington, MA, USA). The Mouse Osteoprotegerin enzyme-linked immunosorbent assay (ELISA) Kit was purchased from Biomatik (Wilmington, DE, USA). Human TNFSF11/RANKL/TRANCE (Sandwich ELISA) ELISA Kit was obtained from LSBio (Seattle, WA, USA). Other reagents and solvents were commercially obtained and used as received.

### 3.2. Rheological Measurements of the Gelatin/HA Solution

A mixture of gelatin and HA was dissolved in distilled water (DW) to obtain a fixed concentration of 6.3 *w*/*v*% solution. The weight ratio of gelatin and HA in the mixed solution was 20:1. The rheological properties of gelatin/HA solution were measured using an AR 2000ex rheometer (TA Instruments, New Castle, DE, USA). All measurements were carried out using a cone-plate geometry (diameter 40 mm, angle 1°). The thermo-responsive behavior of the gelatin/HA solution was measured by investigating the viscosity at a cooling rate of 2 °C/min from 40 °C to 10 °C. The temperature-dependent gelation study was performed to evaluate the srorage modulus (G′) and loss modulus (G″) in oscillatory tests at a cooling rate of 2 °C/min from 40 °C to 10 °C. The oscillation was applied at a frequency of 1 Hz and a 1% strain. The gel point is defined as the temperature at which G′ exceeds G″ during a temperature decrease.

### 3.3. Fabrication of the 3D Composite Scaffolds

The fabrication of composite scaffolds was performed using a 3D printing machine (ROKIT INVIVO, Rokit Healthcare, Seoul, Korea). Before 3D printing the composite scaffolds, gelatin (0.6 g) and HA (0.03 g) were dissolved in 10 mL of DW and mixed with different amounts of HAp nanoparticles. The mixture was stirred vigorously until homogenous pastes were achieved at 40 °C. The contents of HAp nanoparticles in the mixed solutions were 0 (GEHA0), 10 (GEHA10), 20 (GEHA20), and 40 wt% (GEHA40) with respect to the weight of the gelatin and HA. Then, the prepared mixture was loaded into a printing tube equipped with a heating system. The final object model was designed using 3D modeling CAD software (Solidworks, Dassault Systèmes, Vélizy-Villacoublay, France). Uninterrupted composite strands were obtained under the conditions of a nozzle moving speed of 3 mm/s and solution temperature of 25 °C. The composite scaffolds with predesigned morphology (disk-shaped lattice) and dimension (11 mm diameter and 3 mm thickness) were 3D-printed on a low-temperature (−4 °C) plate to maintain the structure during the printing process. The distance between the strands and the porosity in the designed scaffolds were set to 600 μm and 77%, respectively. Simple geometries can be obtained by depositing parallel strands in one layer before changing strand orientation in sequential layers. The common orientation is 0°/90° strand angle in the 3D composite scaffolds. After 3D printing, the 3D composite scaffolds were lyophilized in vacuo.

### 3.4. Crosslinking and Biomineralization of the 3D Composite Scaffolds

The 3D-printed composite scaffolds before crosslinking are water-soluble and mechanically weak. Therefore, a chemical crosslinking process was performed using EDC (40 mM)/NHS (60 mM) solution in 90 *v*/*v*% ethanol to stabilize the structure of the 3D composite scaffolds, followed by lyophilization in vacuo. Then, biomineralization was performed to form apatite crystals on the surface of the 3D composite scaffolds by immersing the prepared scaffolds in a supersaturated calcium/phosphate solution of 3 × SBF (thrice the calcium and phosphate ion concentrations of human plasma) [15]. After 48 h, the composite scaffolds were gently washed with DW to remove unreacted substances, followed by freezing and lyophilization.

The crosslinking degree of the scaffolds was determined using TNBS. To a sample of 3 mg of the scaffold, 1 mL of NaHCO3 solution (pH 8.5) and 1 mL freshly prepared 0.5 *w*/*v*% TNBS solution in DW were added. After the reaction at 40 °C for 2 h, 2 mL of 6N HCl was added. The temperature was raised to 60 °C and maintained for 90 min to solubilize the scaffolds. The resulting solution was diluted with 5 mL of DW, and the absorbance was measured at 345 nm using an ultraviolet–visible spectrometer (U-2900, Hitachi, Japan). The crosslinking degree of the scaffolds was calculated as follows:(1)$$\mathrm{Crosslinking}\;\mathrm{degree}\;\left(\%\right)=\left(1-\frac{absorbance_{s}/mass_{s}}{absorbance_{ncs}/mass_{ncs}}\right)\times 100$$
where the subscripts *s* and *ncs* denote the sample and non-crosslinked sample, respectively.

### 3.5. Surface Morphology of the 3D Composite Scaffolds

The surface morphology of the 3D composite scaffolds was observed using SEM (Mira III, TESCAN, Brno, Czech Republic) after sputter-coating samples with platinum. The topography of the 3D composite scaffold surfaces was analyzed on the basis of AFM measurements acquired with a NanoScope IIIa (Digital Instruments, Bresso, Italy).

### 3.6. Physicochemical and Mechanical Properties of the 3D Composite Scaffolds

The elemental and chemical compositions of the 3D composite scaffolds were measured via energy-dispersive X-ray spectroscopy (EDX) equipped in the SEM and FTIR (ALPHA spectrometer, Bruker Optics, Billerica, MA, USA). FTIR spectra were collected in the wavenumber range from 400 to 4000 cm^−1^ with a resolution of 4 cm^−1^ and 24 scans. The crystalline phases of the 3D composite scaffolds were determined via XRD analysis using a high-resolution X-ray diffractometer (D/MAX-2500V/PC, Rigaku, Akishima, Japan) with Cu Kα radiation. The mechanical properties of the 3D composite scaffolds were identified using a universal testing machine (AGS-X, Shimadzu, Kyoto, Japan) using a force load cell of 10 kN capacity. Each sample (10 × 10 × 3 mm^2^) was tested four times at a loading speed of 1 mm/min, with increasing compression until a 50% strain level was reached. The specific surface area of the composite scaffolds was calculated using Solidworks CAD software, and the pore size was measured using a porosimeter (Autopore V 9620, Micromeritics Instruments, Norcross, GA, USA).

### 3.7. Cell Attachment and Proliferation on the 3D Composite Scaffolds

The bioactivity of the 3D composite scaffolds was evaluated by analyzing cell attachment and proliferation of hBMSCs. All cells were cultured in MEM-α supplemented with 10% FBS and 1% penicillin–streptomycin at 37 °C with 5% CO_2_. Before cell seeding, the 3D composite scaffolds were sterilized with a graded series of ethanol (75%, 50%, and 25%) and 5 h of UV irradiation, rinsed five times each with DPBS, placed into a 48-well tissue culture plate, and fixed with a glass ring (8 mm inner diameter). Subsequently, hBMSCs were placed onto the sterilized composite scaffolds in culture medium at densities of 2 × 10^5^ cells per well, and they were cultured for 7 and 14 days. The cell proliferation of hBMSCs on the 3D composite scaffolds was determined using the MTT protocol at each time point. After culturing, the culture medium was removed, and then 0.2 mL of the MTT solution (5 mg/mL in DPBS) was added to each well containing hBMSCs and incubated for another 4 h. The supernatant was removed, followed by the addition of 0.5 mL DMSO to solubilize the precipitated formazan crystals. Finally, 0.1 mL triplicates from each sample were transferred to a 96-well plate, and the absorbance at 570 nm was determined using an OPSYS-MR microplate reader (Dynex Technology Inc., Chantilly, VA, USA).

The attachment status of hBMSCs cultured on the 3D composite scaffolds was observed using CLSM. hBMSCs (2 × 10^5^ cells per well) were seeded onto the composite scaffolds and cultured for 7 days. Next, cells were fixed with 4% paraformaldehyde for 15 min and rinsed three times using DPBS. Fixed samples were treated with 0.1 *w*/*v*% Triton-X 100 for 5 min to permeabilize the cell membrane, and then they were treated with 1 *w*/*v*% bovine serum albumin for 30 min. The cultured cells were stained with tetramethylrhodamine-conjugated phalloidin (phalloidin-TRITC) and 4′,6-diamidine-2′-phenylindole (DAPI) at room temperature. Finally, the cells were observed using an inverted LSM 700 confocal laser scanning microscope (Carl Zeiss, Oberkochen, Germany).

### 3.8. ALP Activity

The osteogenic differentiation of hBMSCs on the 3D composite scaffolds was evaluated using the ALP activity. ALP activity as an early marker of osteogenic differentiation of hBMSCs was measured using a QuantiChrom Alkaline Phosphatase Assay Kit (BioAssay Systems, Hayward, CA, USA) after 14 days of culture [29]. Based on the hydrolysis of *p*-nitrophenyl phosphate to *p*-nitrophenol by ALP, the absorbance of *p*-nitrophenol was determined at 405 nm as the ALP activity using a microplate reader.

### 3.9. Expression of OPG and RANKL

An OPG is an osteoclastogenesis inhibitory factor that reduces the rate of osteoclastic bone resorption [28]. However, RANKL binds to osteoclasts to activate bone resorption. The levels of OPG and RANKL expression in hBMSCs cultured on the 3D composite scaffolds were determined by measuring the absorbance at 450 nm using Mouse Osteoprotegerin ELISA Kit and Human TNFSF11/RANKL/TRANCE (Sandwich ELISA) ELISA Kit after culturing for 14 days. The concentrations of OPG and RANKL were calculated by linear interpolation of the standard curve. The kits were used according to manufacturers’ specifications.

### 3.10. qRT-PCR Analysis

To confirm the osteogenic differentiation of hBMSCs on the 3D composite scaffolds, the expression levels of osteogenesis-related genes such as COL1, RUNX2, and OPN were determined via qRT-PCR. hBMSCs (2 × 10^5^ cells per well) were seeded onto the composite scaffolds, cultured for 14 days, and transferred into a 15 mL plastic tube containing 2 mL of Trizol solution (Bio Science Technology, Daegu, Korea) to extract RNA. Next, RNA samples were reverse-transcribed into cDNA using PrimeScript RT reagent Kit (Takara, Kusatsu, Japan) according to the manufacturer’s protocol. The composite scaffolds were analyzed using GoTaq^®^ qPCR and RT-qPCR Systems (Promega, Madison, WI, USA). The relative expression level for each target gene was normalized to the expression level of the housekeeping gene glyceraldehyde-3-phosphate dehydrogenase (GAPDH). Table 1 presents all primer sequences used. At least six species per sample were tested.

### 3.11. Statistical Analysis

All data were expressed as the mean value ± standard deviation. Statistical comparison for two different samples was analyzed with one-way analysis of variance followed by Tukey’s test using SigmaPlot 13.0 (Systat Software Inc., San Jose, CA, USA). Significant differences were defined for values of *p* ˂ 0.05.

## 4. Conclusions

In this study, a novel biomimetic fabrication technique was developed to prepare biomineralized gelatin/HA/HAp composite scaffolds for bone tissue engineering. Bone-like tissue engineering scaffolds with adjustable composition and morphology were created by combining the biomimetic approach with 3D printing. The scaffolds obtained had interconnected microporous structures of strands. The biomineralization and morphological characteristics of the scaffolds influenced the physicochemical properties and the biological activity of scaffolds, as well as the bone-forming ability of cells cultured on the composite scaffolds. In particular, biomineralized composite scaffolds, such as GEHA20-ZZS and GEHA20-45S, promoted cell proliferation and osteogenic differentiation more efficiently than other scaffolds. Conclusively, our simple method provides a promising technology for surface modification and geometrical configuration change to improve scaffold osseointegration, resulting in the development of potential new bone graft substitutes.

## Figures and Tables

**Figure 1 ijms-22-06794-f001:**
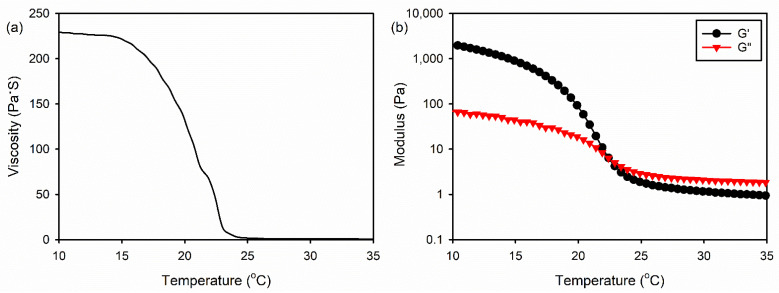
(**a**) Viscosity and (**b**) modulus of the gelatin/HA aqueous solution at various temperatures.

**Figure 2 ijms-22-06794-f002:**
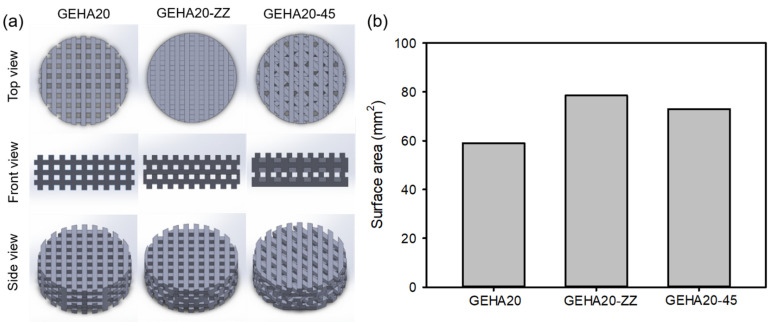
(**a**) Final scaffold models designed using 3D modeling CAD software and (**b**) surface area of scaffold models calculated using 3D modeling CAD software.

**Figure 3 ijms-22-06794-f003:**
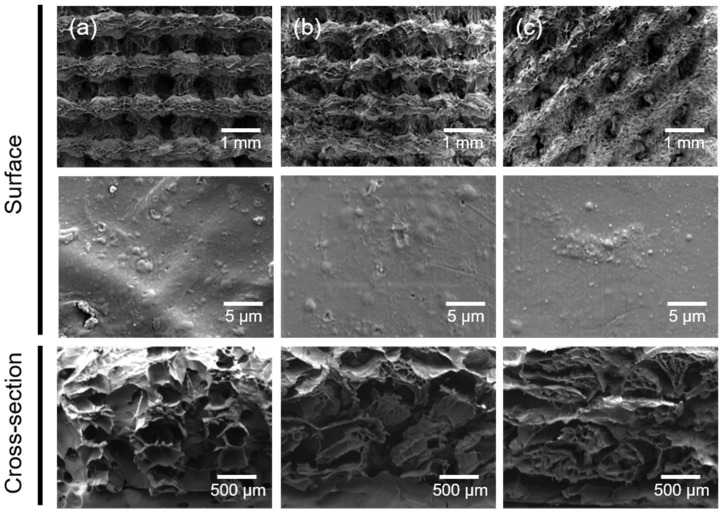
SEM images of (**a**) GEHA20, (**b**) GEHA20-ZZ, and (**c**) GEHA20-45 scaffolds before biomineralization.

**Figure 4 ijms-22-06794-f004:**
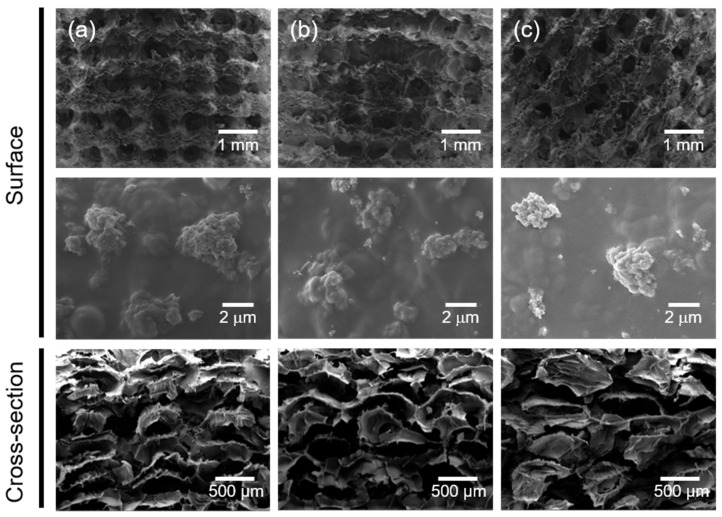
SEM images of (**a**) GEHA20S, (**b**) GEHA20-ZZS, and (**c**) GEHA20-45S composite scaffolds after biomineralization.

**Figure 5 ijms-22-06794-f005:**
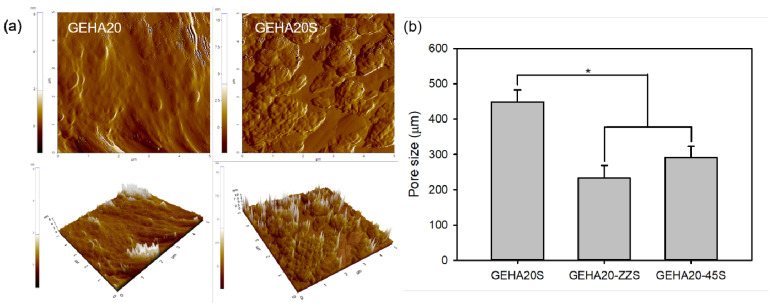
(**a**) AFM micrographs and (**b**) pore size (*n* = 4) of composite scaffolds. * *p* ˂ 0.05 for comparison between two treatment groups.

**Figure 6 ijms-22-06794-f006:**
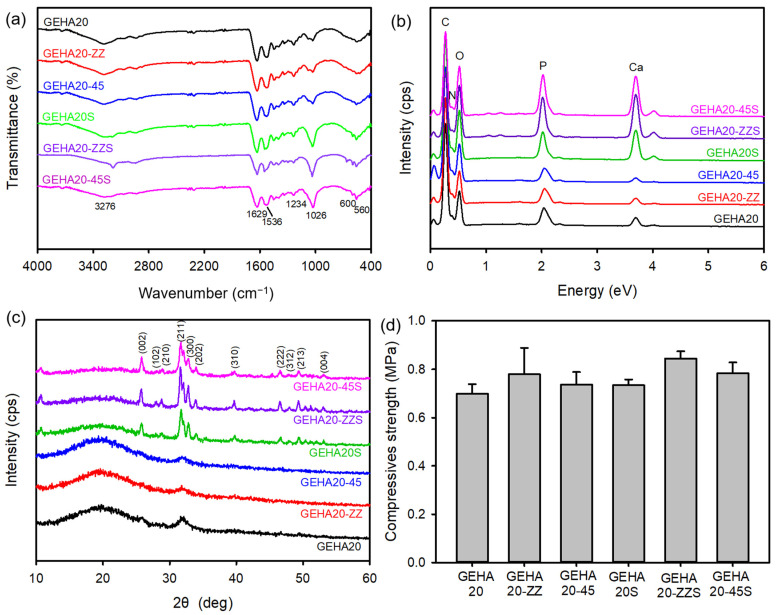
(**a**) FTIR, (**b**) EDX, (**c**) XRD spectra, and (**d**) compressive strength of composite scaffolds (*n* = 4).

**Figure 7 ijms-22-06794-f007:**
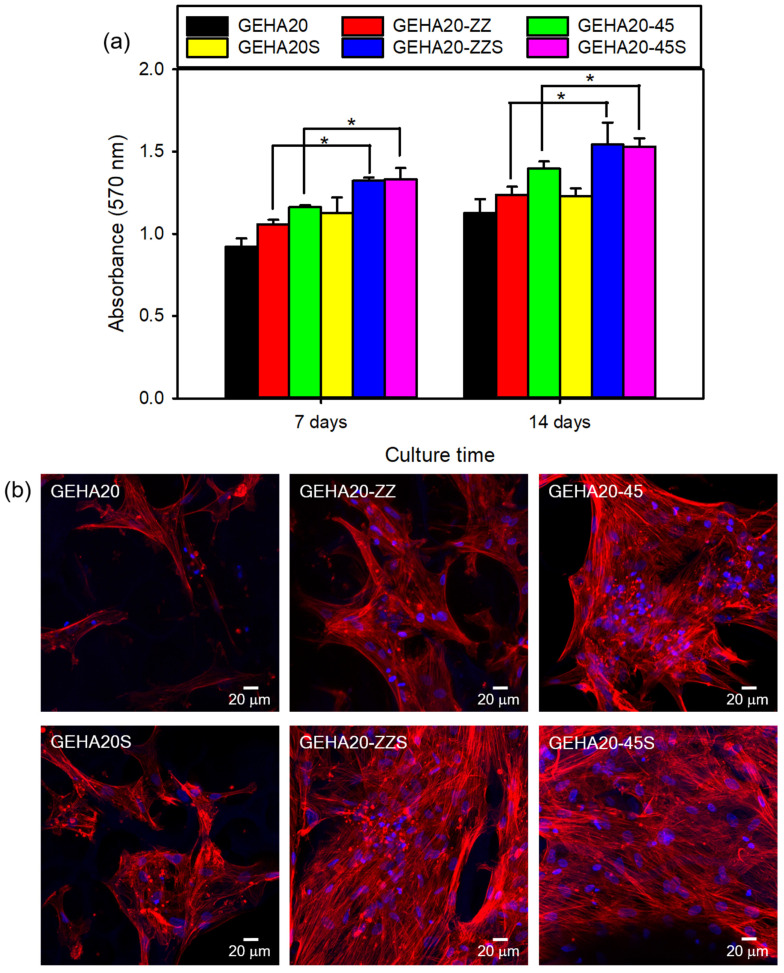
(**a**) Proliferation of hBMSCs cultured on composite scaffolds (*n* = 5) and (**b**) CLSM images of hBMSCs grown on composite scaffolds after culturing for 7 days. * *p* ˂ 0.05 for comparison between two treatment groups.

**Figure 8 ijms-22-06794-f008:**
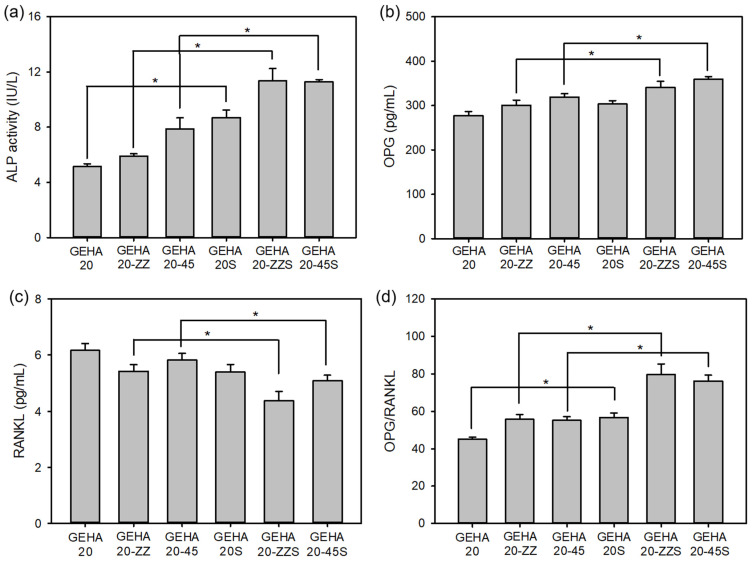
ALP activity and expressions of OPG and RANKL proteins in hBMSCs on the composite scaffolds after 14 days of cell culture (*n* = 5); (**a**) ALP activity, (**b**) OPG expression, (**c**) RANKL expression, and (**d**) OPG/RANKL ratio. * *p* ˂ 0.05 for comparison between two treatment groups.

**Figure 9 ijms-22-06794-f009:**
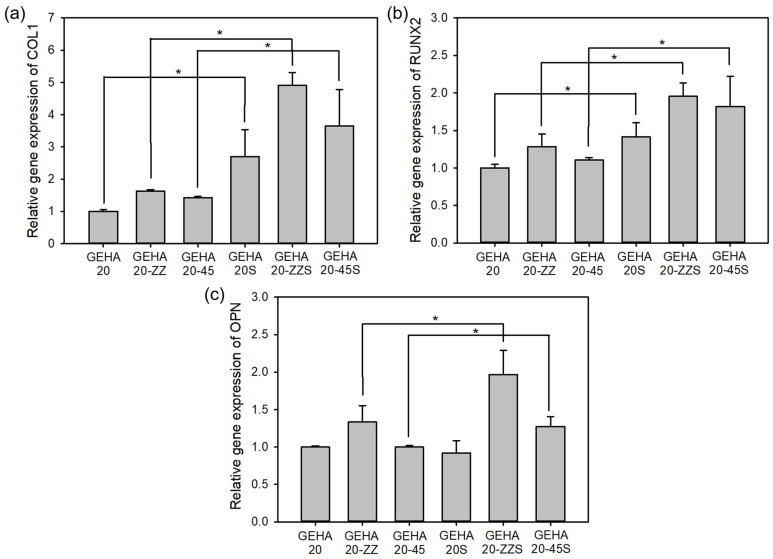
Gene expressions in osteogenic differentiation of hBMSCs cultured on composite scaffolds for 14 days (*n* = 5); (**a**) COL1, (**b**) RUNX2, and (**c**) OPN. * *p* ˂ 0.05 for comparison between two treatment groups.

**Table 1 ijms-22-06794-t001:** qRT-PCR primer sequences for indicated genes.

Gene	Primer Sequence (5′-3′)	
	Forward	Reverse
COL1	TCTAGACATGTTCAGCTTTGTGGAC	TCTGTACGCAGGTGATTGGTG
RUNX2	CACTGGCGCTGCAACAAGA	CATTCCGGAGCTCAGCAGAATAA
OPN	TCACCAGTCTGATGAGTCTCACCATTC	TAGCATCAGGGTACTGGATGTCAGGTC
GAPDH	AGATCATCAGCAATGCAATGCCTCC	ATGGCATGGACTGTGGTCAT

## Data Availability

The data presented in this study are available on request from the corresponding author.

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
