# Peer review of "Effect of Morphological Characteristics and Biomineralization of 3D-Printed Gelatin/Hyaluronic Acid/Hydroxyapatite Composite Scaffolds on Bone Tissue Regeneration"

_ijms, 2021, doi:10.3390/ijms22136794_

Round 1

Reviewer 1 Report

The authors of the manuscript focused on  biomimetic composite scaffolds using a 3D printing method and subsequent biomineralization for improved regeneration. Composite scaffolds are key methods nowadays in biomedical engineering. The most important finding in this study was that biomineralization has a great influence on the biological activity of cells.

The authors in the introduction describe reconstruction of damaged bone defects, here it is necessary to state that the recurrence of bleeding to the joints in congenital bleeding disorders such as hemophilia and congenital afibrinogenemia. Such a case has been reported in a patient with afibrinogenemia and recurrent joint bleeding. Authors should cite this manuscript: ,, Simurda T et al. Perioperative Coagulation Management in a Patient with Congenital Afibrinogenemia during Revision Total Hip Arthroplasty. Semin Thromb Hemost. 2016 Sep;42(6):689-92. doi: 10.1055/s-0036-1585079“.

The methodical part is extensive and precisely written. I would like to commend the use of individual laboratory methods.

Tables and figures in the text are very clearly written.

I have to say that with these 24 references there are 21 references newer than 5 years old.

Author Response

Reviewer 1

The authors of the manuscript focused on biomimetic composite scaffolds using a 3D printing method and subsequent biomineralization for improved regeneration. Composite scaffolds are key methods nowadays in biomedical engineering. The most important finding in this study was that biomineralization has a great influence on the biological activity of cells.

Point 1: The authors in the introduction describe reconstruction of damaged bone defects, here it is necessary to state that the recurrence of bleeding to the joints in congenital bleeding disorders such as hemophilia and congenital afibrinogenemia. Such a case has been reported in a patient with afibrinogenemia and recurrent joint bleeding. Authors should cite this manuscript: ,, Simurda T et al. Perioperative Coagulation Management in a Patient with Congenital Afibrinogenemia during Revision Total Hip Arthroplasty. Semin Thromb Hemost. 2016 Sep;42(6):689-92. doi: 10.1055/s-0036-1585079“.

  • Response 1: According to the comment of the reviewer, we added one sentence to explain bleeding disorders in page 1 and also cited provided one article as reference 2. Thank you for your kind comment.

Point 2: The methodical part is extensive and precisely written. I would like to commend the use of individual laboratory methods.

  • Response 2: As pointed out by the reviewer, we rewrote “3. Materials and Methods” part as individual laboratory methods.

Point 3: Tables and figures in the text are very clearly written.

  • Response 3: Thank you for your kind comment.

Point 4: I have to say that with these 24 references there are 21 references newer than 5 years old.

  • Response 4: We tried to refer to recent results for fabrication of 3D composite scaffolds and their applications in bone tissue regeneration. Thus many of cited references in this manuscript are most recent papers published after 2017.

Thank you for reviewing our paper.

Reviewer 2 Report

The manuscript describes the Effect of Morphological Characteristics and Biomineralization of 3D-Printed Composite Scaffolds on Bone Tissue Regeneration. The manuscript focuses an interesting subject and the techniques were well used. The composites scaffolds are well characterized and present a very promising behaviour to be used as new bone substitutes. The data are in most cases well presented and discussed. Some corrections are needed:

  • Page 3: when authors refer the crosslinking degrees of the scaffolds it must be said how it was calculated. It is not enough saying just it was according to reference 17. The procedure must be briefly described.
  • When discussion AFM results (page 5) it is presented the Ra values for GEHA20 before and after biomineralization indicating an increase. And for the other samples? Was it observed an identical behaviour? It must be state in the text.
  • Page 7: why just present EDX and XRD results for the sample GEHA20 (Figure 5b) if authors have said in the previous page that the other 2 samples are the one which have suitable pore sizes for bone tissue engineering? It would make more sense present data from the other samples, or, at least discuss that on the manuscript.
  • Also on page 7, please check the template manuscript.
  • On page 9, it would facilitate the reading if authors can group colours by the same sample before and after mineralization (black/yellow; red/blue; green/mangenta).
  • Fig 6: absorbance results at 7 and 14 days are presented at Fig 6a. Why on Fig 6b images are from 10 days of culture?
  • The mention to Fig 6b on the text must appear before the figure itself. The same for Fig 7.
  • Page 12: despite have been described elsewhere in the manuscript, please describe again hBMSCs.
  • Page 13, line 287 and 290: please indicate the temperature at which the mixture was stirred and at was printed using the heating system.
  • Section 3.3: “as-3D-printed” – please clarify
  • Section 3.4: Please indicate: - number of runs, resolution and the support of samples in FTIR analysis; - Mechanical properties: which were the dimensions of the samples? Standard used?
  • Section 3.5: was made any blank assay using the samples with medium without cells and incubating at the same conditions? And a positive control? I.e., medium and cells without samples)? Are the absorbance values presented before the result of the result of subtracting these blank results or the absorbance value direct read?
  • Conclusions: Since there are no “in vivo” the final sentence must be reformulated. For instance, “Development of a potential new bone graft substitute”.

Author Response

Reviewer 2

The manuscript describes the Effect of Morphological Characteristics and Biomineralization of 3D-Printed Composite Scaffolds on Bone Tissue Regeneration. The manuscript focuses an interesting subject and the techniques were well used. The composites scaffolds are well characterized and present a very promising behaviour to be used as new bone substitutes. The data are in most cases well presented and discussed. Some corrections are needed:

Point 1: Page 3: when authors refer the crosslinking degrees of the scaffolds it must be said how it was calculated. It is not enough saying just it was according to reference 17. The procedure must be briefly described.

  • Response 1: As pointed out by the reviewer, we added several sentences to explain the procedure of crosslinking degree analysis of the scaffolds in “3.4. Crosslinking and Biomineralization of the 3D Composite Scaffolds” section in page 17. Thank you for kind comment.

Point 2: When discussion AFM results (page 5) it is presented the Ra values for GEHA20 before and after biomineralization indicating an increase. And for the other samples? Was it observed an identical behaviour? It must be state in the text.

  • Response 2: According to the comment of the reviewer, we added the change of the Ra values for GEHA20-ZZ and GEHA20-45 before and after biomineralization in page 7. The change of the Ra values for GEHA20-ZZ and GEHA20-45 before and after biomineralization was almost identical to the change of the Ra value for GEHA20.

Point 3: Page 7: why just present EDX and XRD results for the sample GEHA20 (Figure 5b) if authors have said in the previous page that the other 2 samples are the one which have suitable pore sizes for bone tissue engineering? It would make more sense present data from the other samples, or, at least discuss that on the manuscript.

  • Response 3: As pointed by the reviewer, we retried the measurements of EDX and XRD for all samples and the results were added in Figure 6 (Figure 5 changed to Figure 6).

Point 4: Also on page 7, please check the template manuscript.

  • Response 4: We checked the template manuscript and corrected the template. Thank you for kind advice.

Point 5: On page 9, it would facilitate the reading if authors can group colours by the same sample before and after mineralization (black/yellow; red/blue; green/mangenta).

  • Response 5: According to the comment of the reviewer, we changed group colors by the same samples before and after biomineralization in Figure 7 (Figure 6 changed to Figure 7).

Point 6: Fig 6: absorbance results at 7 and 14 days are presented at Fig 6a. Why on Fig 6b images are from 10 days of culture?

  • Response 6: Thank you for pointing out our mistake. The CLSM images of hBMSCs grown on composite scaffolds were observed after culturing for 7 days. Therefore, we changed “10 days” to “7 days” in figure caption and in the text.

Point 7: The mention to Fig 6b on the text must appear before the figure itself. The same for Fig 7.

  • Response 7: As commented by the reviewer, the mentions (explanations) for Figures 7b and 8b-8d were transferred to before the Figures themselves (Figures 6b and 7b-7d changed to Figures 7b and 8b-8d).

Point 8: Page 12: despite have been described elsewhere in the manuscript, please describe again hBMSCs.

  • Response 8: According to the comment of the reviewer, we described again full name of hBMSCs in page 16.

Point 9: Page 13, line 287 and 290: please indicate the temperature at which the mixture was stirred and at was printed using the heating system.

  • Response 9: According to the comment of the reviewer, we indicated the temperatures for mixing solutions and printing scaffolds in “3.3. Fabrication of the 3D Composite Scaffolds” section, in page 17. The mixture was stirred vigorously until homogenous pastes were achieved at 40 °C. In addition, the temperature (25 °C) of printing solution was maintained during the 3D printing process.

Point 10: Section 3.3: “as-3D-printed” – please clarify

  • Response 10: As pointed out by the reviewer, we clarified “as-3D-rpinted”. We changed “The as-3D-printed composite scaffolds” to “The 3D-printed composite scaffolds before crosslinking”.

Point 11: Section 3.4: Please indicate: - number of runs, resolution and the support of samples in FTIR analysis; - Mechanical properties: which were the dimensions of the samples? Standard used?

  • Response 11: As commented by the reviewer, we added one sentence to indicate number of scans, resolution, and wavenumber range in FTIR analysis. In addition, the dimension of samples was added in mechanical property test in “3.6. Physicochemical and Mechanical Properties of the 3D Composite Scaffolds” section.

Point 12: Section 3.5: was made any blank assay using the samples with medium without cells and incubating at the same conditions? And a positive control? I.e., medium and cells without samples)? Are the absorbance values presented before the result of the result of subtracting these blank results or the absorbance value direct read?

  • Response 12: The absorbance values in “3.7. Cell Attachment and Proliferation on the 3D Composite Scaffolds” section were obtained by subtracting the blank’s value (using the samples with medium and without cells) from the samples’ values (with cells).

Point 13: Conclusions: Since there are no “in vivo” the final sentence must be reformulated. For instance, “Development of a potential new bone graft substitute”.

  • Response 13: According to the comment of the reviewer, we changed the last sentence in Conclusions to “resulting in the development of potential new bone graft substitutes”.

Thank you for reviewing our paper.

Reviewer 3 Report

Although the part dealing with cell studies is well written and presented the first part of the manuscript is weak. My suggestion is major revision

  • The title and the abstract are too general without mentioning the materials used. The authors used “gelatin based” gels to print 3D scaffolds. There is no reference about gelatin neither in the title nor in the abstract.
  • The authors used one concentration of hydroxyapatite. Please explain
  • Articles dealing with 3D printing of gels usually contain information about the rheology of the gels. Not such results are presented.
  • To my opinion a control 3D printed gel experiment without the presence of Hap crystals should be also presented.
  • Magnification of Figure 3 is low and the characteristic shape of hydroxyapatite crystals is not visible.
  • Figure 5c. Replace the stars symbol with the Miller indices
  • Lines 176-182 . Discuss the results of the mechanical properties with the literature. For example,
  • with other gels or materials used in bone tissue engineering

Author Response

Reviewer 3

Although the part dealing with cell studies is well written and presented the first part of the manuscript is weak. My suggestion is major revision

Point 1: The title and the abstract are too general without mentioning the materials used. The authors used “gelatin based” gels to print 3D scaffolds. There is no reference about gelatin neither in the title nor in the abstract.

  • Response 1: As commented by the reviewer, we added the materials’ name (gelatin/hyaluronic acid/hydroxyapatite) used in this study in the title and abstract.

Point 2: The authors used one concentration of hydroxyapatite. Please explain

  • Response 2: We actually used four concentrations of hydroxyapatite to fabricate the composite scaffolds, such as 0 (GEHA0), 10 (GEHA10), 20 (GEHA20), and 40 wt% (GEHA40) with respect to the weight of the gelatin and hyaluronic acid. The significant changes of morphology, chemical structure, and mechanical property were not observed in the composite scaffolds containing different amount of hydroxyapatite nanoparticles. Thus we used one concentration of hydroxyapatite (GEHA20). According to the comment of the reviewer, we added Figures S1 and S2 concerning the morphology, chemical structure, and mechanical property of GEHA composite scaffolds containing different amount of hydroxyapatite nanoparticles in the supplementary information. We also added one paragraph in page 3 to explain the morphology, chemical structure, and mechanical property of various GEHA composite scaffolds (Figures S1 and S2).

Point 3: Articles dealing with 3D printing of gels usually contain information about the rheology of the gels. Not such results are presented.

  • Response 3: As pointed out by the reviewer, we measured the viscosity and rheological behavior of gelatin/hyaluronic acid solution, and the results were exhibited in Figure 1. In addition, we added some sentences in pages 3 and 17 to explain the experimental methods and results on the viscosity and rheological behavior of gelatin/hyaluronic acid solution.

Point 4: To my opinion a control 3D printed gel experiment without the presence of Hap crystals should be also presented.

  • Response 4: As mentioned above (in Response 2), we added analysis results of the 3D gelatin/hyaluronic acid scaffold fabricated without HAp nanoparticles as control in Figures S1 and S2.

Point 5: Magnification of Figure 3 is low and the characteristic shape of hydroxyapatite crystals is not visible.

  • Response 5: As commented by the reviewer, we changed the SEM images to high magnification images to observe the characteristic shape of apatite crystals after biomineralization in Figure 4 (Figure 3 changed to Figure 4).

Point 6: Figure 5c. Replace the stars symbol with the Miller indices

  • Response 6: According to the comment of the reviewer, we replaced the star symbol with the Miller indices in Figure 6c (XRD spectra, Figure 5 changed to Figure 6))

Point 7: Lines 176-182 . Discuss the results of the mechanical properties with the literature. For example, with other gels or materials used in bone tissue engineering.

  • Response 7: As pointed by the reviewer, we discussed the results of the mechanical properties with the literatures in page 8. These literatures were added as references to discuss the mechanical properties of the composite scaffolds in refs. 25 and 26.

Thank you for reviewing our paper.

Round 2

Reviewer 1 Report

The presented manuscript has been corrected in response to the suggestions. The authors have followed the recommendations of the reviewer. After the revision, the provided data and  addition of the results became more clear.  I would like to thank the authors for resubmitting the manuscript and explaining the obscure points from the previous version.

Reviewer 2 Report

The manuscript was successfully revised accordingly reviewers’ suggestions. Just a word needs text editing:

 Page 3, line 101: it should be thermo-responsive instead of thermos-responsive

Reviewer 3 Report

The revised manuscript has significantly been improved